# OT Score: An OT based Confidence Score for Source Free Unsupervised Domain Adaptation

## Abstract

We address the computational and theoretical limitations of current distributional alignment methods for source-free unsupervised domain adaptation (SFUDA). In particular, we focus on estimating classification performance and confidence in the absence of target labels. Current theoretical frameworks for these methods often yield computationally intractable quantities and fail to adequately reflect the properties of the alignment algorithms employed. To overcome these challenges, we introduce the Optimal Transport (OT) score, a confidence metric derived from a novel theoretical analysis that exploits the flexibility of decision boundaries induced by Semi-Discrete Optimal Transport alignment. The proposed OT score is intuitively interpretable and theoretically rigorous. It provides principled uncertainty estimates for any given set of target pseudo-labels. Experimental results demonstrate that OT score outperforms existing confidence scores. Moreover, it improves SFUDA performance through training-time reweighting and provides a reliable, label-free proxy for model performance.

## 1 Introduction

In recent years, deep neural networks have achieved remarkable breakthroughs across a wide range of applications. However, if the distribution of the training and test data differs, significant performance degradation occurs, which is known as a domain shift (Tsymbal, 2004), which makes retraining critical for the model to re-gain the generalization ability in new domains.

Unsupervised domain adaptation (UDA) mitigates the domain shift problem where only unlabeled data is accessible in the target domain (Glorot et al., 2011). A key approach for UDA is aligning the distributions of both domains by mapping data to a shared latent feature space. Consequently, a classifier trained on source domain features in this space can generalize well to the target domain. Several existing works (Long et al., 2015; 2017; Damodaran et al., 2018; Courty et al., 2016; Rostami & Galstyan, 2023) exhibit a principled way to transform target distribution to be "closer" to the source distribution so that the classifier learned from the source data can be directly applied to the target domain thus pseudo-labels (or predictions) can be made accordingly.

This leads to the question of whether such transformations from the target to the source distribution can accurately match the corresponding class-conditional distributions. For any given target dataset, it is always possible to align its feature distribution with that of the source domain using a divergence function, regardless of whether classes overlap. However, performing UDA in this way is reasonable only if target features remain well-separated by the decision boundaries induced through alignment in the latent feature space—something that is typically difficult to determine in practice. Moreover, the marginal distribution alignment approach complicates the identification of samples with low-confidence pseudo-labels (i.e., samples close to overlapping regions), potentially causing noisy supervision and thus degrading classification performance. This issue becomes particularly critical when no labeled information for the target data is available. Some existing works (Luo & Ren, 2021; Ge et al., 2023; Le et al., 2021) minimize a class-conditional discrepancy between the class-conditional feature distributions $P_{\mathcal{S}}(Z \mid Y)$ and $P_{\mathcal{T}}(Z \mid Y)$. However, using pseudo labels from model predictions to determine the target class-conditional distributions exposes the alignment to noisy supervision—especially early in training.

Under the Optimal Transport (OT) framework, it has been investigated in some theoretical works that the generalization error on the target domain is controlled by both the marginal alignment loss and

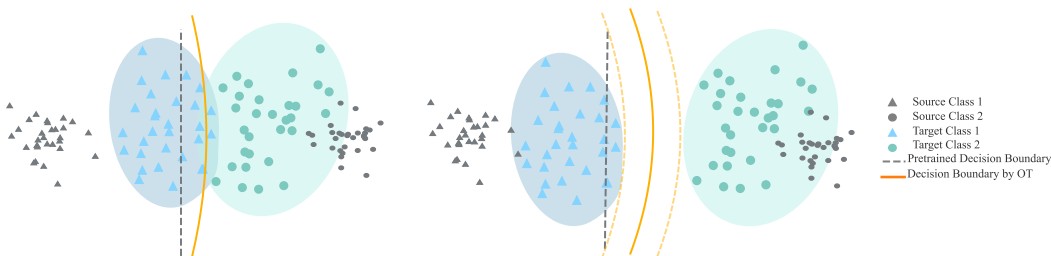

Figure 1: (Left) Overlapping clusters. (Right) Separated clusters with flexible decision boundaries.

the entanglement between the source and target domains. For example, Redko et al. (2017) proves the following:

**Theorem 1** (Informal Redko et al. (2017)). *Under certain assumptions, with probability at least $1 - \delta$ for all hypothesis $h$ and $\varsigma' < \sqrt{2}$ the following holds:*

$$\epsilon_{\mathcal{T}}(h) \leq \epsilon_{\mathcal{S}}(h) + W_1\left(\hat{\mu}_{\mathcal{S}}, \hat{\mu}_{\mathcal{T}}\right) + \sqrt{2\log\left(\frac{1}{\delta}\right)/\varsigma'}\left(\sqrt{\frac{1}{N_{\mathcal{S}}}} + \sqrt{\frac{1}{N_{\mathcal{T}}}}\right) + \lambda$$

*where $\lambda$ is the combined error of the ideal hypothesis $h^*$ that minimizes the combined error of $\epsilon_{\mathcal{S}}(h) + \epsilon_{\mathcal{T}}(h)$.*

A bad pulling strategy on target domain $\mathcal{T}$ might minimize $W_1$ term to $0$ without any guarantee for the $\lambda$ term in the feature space. Similarly, Koç et al. (2025) also show that, during the optimal transport association process, the source inputs $x$ can be associated to target inputs $x'$ that have different matching labels. Minimizing the marginal Wasserstein distance between such entangled pairs can cause the entanglement term to increase. To address these challenges, we will focus on the following question in this work:

**Question: What is the condition on the domain shift so that the target distribution can be aligned back to the source while preserving the correct class labels? Additionally, with only potentially noisy target pseudo-labels available, is there a theoretically guaranteed and computable metric to quantify the degree of violation of this condition?**

Formally, we seek conditions under which the OT between the marginals is label-preserving—i.e., it decomposes into per-class OT between the class-conditional marginals. We formalize and prove these conditions in Section 3. Guided by our theoretical analysis under the semi-discrete OT framework (Section 3.2), we propose the **OT score**—a confidence metric designed to quantify uncertainty in pseudo-labeled target samples. It measures the degree to which the assigned pseudo-label would violate marginal alignment, thereby serving as a diagnostic of class-conditional alignment. As illustrated in Figure 1, the OT score reflects the flexibility of decision boundaries induced by semi-discrete OT alignment, which enables effective uncertainty estimation in the target domain. This allows the algorithm to abstain from classifying samples with high uncertainty. Compared to fully continuous or fully discrete OT formulations, semi-discrete OT is computationally more efficient, especially in high-dimensional spaces and large-scale datasets. A detailed comparison with existing confidence scores is provided in Appendix A.

We also propose two applications of OT score. First, within SFUDA it acts as a training-time reweighting signal: less confident pseudo labels are down-weighted, suppressing harmful updates and improving accuracy. Second, it provides a reliable label-free proxy for target performance: the mean OT score serves as a surrogate for target error, enabling model selection without target labels.

**Contributions:**

- We provide theoretical justifications about allowed distribution shifts in order to have a label-preserving OT.

- We define a novel confidence score, the OT score, which is theoretically interpretable and accounts for the geometry induced by OT alignment between the source and target distributions.

- Experimental results demonstrate that filtering out low-confidence predictions consistently improves classification accuracy, and that the proposed OT score significantly outperforms existing confidence metrics across diverse pseudo-labeling strategies.

- We demonstrate two practical uses of the OT score: (i) as a training-time reweighting signal for SFUDA that down-weights less confident target pseudo-labels to suppress harmful updates and improve accuracy; and (ii) as a label-free proxy for target performance, which enables model selection without target labels.

**Notation**. Given any probability measure $\mu$ and a measurable map $T$ between measurable spaces, $T : \mathcal{X} \longrightarrow \mathcal{Y}$, we denote $T_{\#}\mu$ the pushforward measure on $\mathcal{Y}$ which is characterized by $(T_{\#}\mu)(A) = \mu(T^{-1}(A))$ for measurable set $A$. Let $\hat{\mu}$ denote the corresponding empirical measure $\frac{1}{N}\sum_{i=1}^{N}\delta_{x_i}$ where $x_i$ are i.i.d. samples from $\mu$. We also write $x \in \hat{\mu}$ to indicate $x \in \{x_i\}_{i=1}^{N}$. If not otherwise specified, $\|\cdot\|$ represents the Euclidean norm.

## 2 OPTIMAL TRANSPORT AND DOMAIN ADAPTATION

In this section, we first present the domain adaptation problem. Then we give necessary backgrounds of optimal transport.

### 2.1 DOMAIN ADAPTATION

Let $\Omega \subseteq \mathbb{R}^d$ be the sample space and $\mathcal{P}(\Omega)$ be the set of all probability measures over $\Omega$. In a general supervised learning paradigm for classification problems, we have a labeling function $f_{\boldsymbol{\theta}^*} : \mathbb{R}^d \to \mathbb{R}^k$ obtained from a parametric family $f_{\boldsymbol{\theta}}$ by training on a set of points $\mathbf{X}^{\mathcal{S}} = \{x_1^{\mathcal{S}}, ..., x_{N^{\mathcal{S}}}^{\mathcal{S}}\}$ sampled from a source distribution $P_{\mathcal{S}} \in \mathcal{P}(\Omega)$ and corresponding one-hot encoded labels $\mathbf{Y}^{\mathcal{S}} = \{y_1^{\mathcal{S}}, ..., y_{N^{\mathcal{S}}}^{\mathcal{S}}\}$.

Let $\mathbf{X}^{\mathcal{T}} = \{x_1^{\mathcal{T}}, ..., x_{N^{\mathcal{T}}}^{\mathcal{T}}\}$ be a dataset sampled from a target distribution $P_{\mathcal{T}} \in \mathcal{P}(\Omega)$ without label information. The difference between $P_{\mathcal{S}}$ and $P_{\mathcal{T}}$ may lead to a poor performance if we use $f_{\boldsymbol{\theta}^*}$ for the new classification problem. In order to overcome the challenge of distributional shift, a common way is to decompose a neural network $f_{\boldsymbol{\theta}}$ into a feature mapping $\phi_{\boldsymbol{v}}$ composed with a classifier $h_{\boldsymbol{w}}$ such that $f_{\boldsymbol{\theta}} = h_{\boldsymbol{w}} \circ \phi_{\boldsymbol{v}}$, followed by minimizing the distance between $(\phi_{\boldsymbol{v}^*})_{\#}P_{\mathcal{S}}$ and $(\phi_{\boldsymbol{v}})_{\#}P_{\mathcal{T}}$ so that the target distribution will be aligned with the source distribution in the feature space. Then we may classify target data points based on the optimization result in the feature space. Various choices of divergence objective $D((\phi_{\boldsymbol{v}^*})_{\#}P_{\mathcal{S}}, (\phi_{\boldsymbol{v}})_{\#}P_{\mathcal{T}})$ can be utilized. In this work, we focus on the distributional alignment between $(\phi_{\boldsymbol{v}^*})_{\#}P_{\mathcal{S}}$ and $(\phi_{\boldsymbol{v}})_{\#}P_{\mathcal{T}}$ using Wasserstein distance.

### 2.2 OPTIMAL TRANSPORT

#### 2.2.1 GENERAL THEORY OF OT

Given two probability distributions $\mu, \nu \in \mathcal{P}(\mathbb{R}^d)$, the Wasserstein-$p$ distance for $p \in [1, +\infty]$ is defined by

$$W_p(\mu, \nu) := \left( \min_{\gamma \in \Gamma(\mu, \nu)} \int_{\mathbb{R}^d \times \mathbb{R}^d} \|x - y\|^p d\gamma \right)^{\frac{1}{p}},$$

where $\Gamma(\mu, \nu)$ is the collection of all couplings of $\mu$ and $\nu$. The optimization problem

$$\min_{\gamma \in \Gamma(\mu, \nu)} \int_{\mathbb{R}^d \times \mathbb{R}^d} \|x - y\|^p d\gamma \tag{KP}$$

is referred as the Kantorovitch problem in optimal transport. It is shown by Kantorovich–Rubinstein Duality theorem that (KP) has a dual form (Santambrogio, 2015):

**Theorem 2** (Kantorovich–Rubinstein Duality)**.**

$$\min_{(KP)} = \sup\Big\{\int_{\mathbb{R}^d} \phi(x)\,d\mu + \int_{\mathbb{R}^d} \psi(y)\,d\nu : (\phi, \psi) \in Lip_b(\mathbb{R}^d) \times Lip_b(\mathbb{R}^d),\ \phi(x) + \psi(y) \leq \|x - y\|^p\Big\}.$$

*In addition, when the supremum in the dual formulation is a maximum, the optimal value is attained at a pair $(\phi, \phi^c)$ with $\phi$, $\phi^c$ bounded and Lipschitz, where $\phi^c(y) := \inf_{x \in \mathbb{R}^d} \|x - y\|^p - \phi(x)$.*

With the dual problem introduced, Brenier (1991) proves Brenier's theorem, which gives a sufficient condition under which the minimizer of the optimal transport problem is unique and is induced by a map $T = \nabla \phi$ for some convex function $\phi$, i.e. the OT map exists.

Under mild conditions on $\mu$ and $\nu$, Brenier's theorem is satisfied when $c(x, y) = \|x - y\|_p$ for $p > 1$. Although there is no guarantee about uniqueness of the optimal transport map when $p = 1$, the existence of an optimal transport map can be proved through a secondary variational problem Santambrogio (2015):

**Theorem 3** (Existence of optimal transport map when $p = 1$)**.** *Let $O(\mu, \nu)$ be the optimal transport plans for the cost $\|x - y\|$ and denote by $K_p$ the functional associating to $\gamma \in \mathcal{P}(\Omega \times \Omega)$, the quantity $\int \|x - y\|^p\,d\gamma$. Under the usual assumption $\mu \ll \mathcal{L}^d$, the secondary variational problem*

$$\min\{K_2(\gamma) : \gamma \in O(\mu, \nu)\}$$

*admits a unique solution $\bar{\gamma}$, which is induced by a transport map T.*

### 2.2.2 SEMI-DISCRETE OPTIMAL TRANSPORT

A special case of interest is when $\nu = \sum_{j=1}^m b_j \delta_{y_j}$ is a discrete probability measure. Adapting the duality result to this setting, we have

$$W_p^p(\mu, \nu) = \max_{\boldsymbol{w} \in \mathbb{R}^m} \int_{\mathbb{R}^d} \boldsymbol{w}^c(x)\,d\mu + \sum_{j=1}^m w_j b_j,$$

and in this case, $\boldsymbol{w}^c(x) := \min_j \|x - y_j\|^p - w_j$.

We can define a disjoint decomposition of the whole space using the Laguerre cells associated to the dual weights $\boldsymbol{w}$:

$$\mathbb{L}_{\boldsymbol{w}}(y_j) := \Big\{x \in \mathbb{R}^d : \forall j' \neq j, \|x - y_j\|^p - w_j \leq \|x - y_{j'}\|^p - w_{j'}\Big\}.$$

Then

$$W_p^p(\mu, \nu) = \max_{\boldsymbol{w} \in \mathbb{R}^m} \sum_{j=1}^m \int_{\mathbb{L}_{\boldsymbol{w}}(y_j)} \big(\|x - y_j\|^p - w_j\big)\,d\mu + \langle \boldsymbol{w}, \boldsymbol{b}\rangle.$$

The optimization problem above can be solved by (stochastic) gradient ascent methods since the $j$-th entry of gradient for the objective function can be computed via $b_j - \int_{\mathbb{L}_{\boldsymbol{w}}(y_j)} d\mu$. Once the optimal vector $\boldsymbol{w}$ is computed, the optimal transport map $T_\mu^\nu$ simply maps $x \in \mathbb{L}_{\boldsymbol{w}}(y_j)$ to $y_j$ Peyré et al. (2019). Also, it can be shown such OT map is unique under mild assumptions (Hartmann & Schuhmacher, 2017; Geiß et al., 2013). In the rest of the paper, for any $x \in \text{supp}\,\mu$ and $y_j \in \text{supp}\,\nu$, we denote $\tilde{d}_{\boldsymbol{w}}(x, y_j) := \|x - y_j\|^p - w_j$. Convergence properties of semi-discrete optimal transport have been studied extensively; see, e.g., Genevay et al. (2016) and Peyré et al. (2019) for details.

## 3 THEORETICAL ANALYSIS

In this section, we present theoretical insights into the use of OT for addressing DA problems. Complete proofs of all theoretical results are provided in Appendix C. For clarity and tractability, we focus on binary classification tasks. An extension to multiclass classification follows by a one-vs-all reduction. As discussed in Section 2.1, our interest lies in neural network–based DA. To this end, we adopt assumptions inspired by Neural Collapse (Kothapalli, 2022), a prevalent phenomenon observed in well-trained neural networks. The extent to which the target feature distribution conforms to the Neural Collapse structure depends on the severity of the distributional shift between the source and target domains.

**Remark 1** (Neural Collapse). *Neural collapse (NC) is a phenomenon observed in well-trained neural networks where the learned features of samples belonging to the same class converge to a single point or form tightly clustered structures in the feature space, while the features of different classes become maximally separated. NC emerges while training modern classification DNNs past zero error to further minimize the loss (Papyan et al., 2020). During NC, the class means of the DNN's last-layer features form a symmetric structure with maximal separation angle, while the features of each individual sample collapse to their class means. This simple structure of the feature layer not only appears beneficial for generalization but also helps in transfer learning and adversarial robustness. There are three main theoretical frameworks proposed to explain the emergence of NC: "Unconstrained Features Model" (Lu & Steinerberger, 2022; Tirer & Bruna, 2022; Ji et al., 2021), "Local Elasticity" (Zhang et al., 2021) and "Neural (Tangent Kernel) Collapse" (Seleznova et al., 2024).*

In the following subsection, we focus on the setting where the class-conditional distributions in both the source and target domains are supported on, or concentrated within, bounded subsets of the feature space. Stronger NC in the source representation yields smaller cluster radii, thereby strengthening our results. Under this assumption, we analyze how data clusters are transported by the OT map.

## 3.1 Sufficient Conditions for Correct Classification

We begin by presenting a necessary condition on the target data distribution under which correct classification can be expected after applying optimal transport. The following theorem quantifies the relationship between the probability of misclassification and the concentration properties of class-conditional distributions. Intuitively, if each class distributions is concentrated within a bounded region and these regions are well-separated across classes, classification results after OT map will be correct with high probability.

**Theorem 4.** *Suppose for each of the probability measures $\mu_i, \nu_i$ there exist disjoint bounded sets $E_{\mu_i}(or\ E_{\nu_i})$ such that $\mu_i(E_{\mu_i}) \geq 1 - \epsilon$ and $(r_{\mu_1} + r_{\nu_1} + l_1) + (r_{\mu_2} + r_{\nu_2} + l_2) < L_1 + L_2$, where $r_{\mu_i}(or\ r_{\nu_i})$ is the diameter of $E_{\mu_i}(or\ E_{\nu_i})$, $l_i = d(E_{\mu_i}, E_{\nu_i})$, $L_1 = d(E_{\mu_1}, E_{\nu_2})$, $L_2 = d(E_{\mu_2}, E_{\nu_1})$. Assume further that $E_{\nu_1}$ and $E_{\nu_2}$ are correctly separated by the trained classifier. Then with probability greater than $1 - 7\epsilon$, target samples will be correctly classified after the optimal transportation $T_\nu^\mu$.*

**Remark 2.** *Our concentration assumption applies to various probability distributions including subgaussian distributions.*

The proof is based on the intuitive observation from the following lemma:

**Lemma 5.** *Suppose we have probability measures $\mu_i$ and $\nu_i$ with bounded support. Also assume $\operatorname{supp} \mu_1$ and $\operatorname{supp} \mu_2$ are disjoint, $\operatorname{supp} \nu_1$ and $\operatorname{supp} \nu_2$ are disjoint. Let $r_{\mu_i}$ denote the diameter of the support of $\mu_i$ and set $l_i = d(\operatorname{supp} \mu_i, \operatorname{supp} \nu_i)$, $L_1 = d(\operatorname{supp} \mu_1, \operatorname{supp} \nu_2)$, $L_2 = d(\operatorname{supp} \mu_2, \operatorname{supp} \nu_1)$. Suppose $\mu := \frac{1}{2}\mu_1 + \frac{1}{2}\mu_2$, $\nu := p\nu_1 + (1-p)\nu_2$ for some $p \in (0, \frac{1}{2}]$. If $(r_{\mu_1} + r_{\nu_1} + l_1) + (r_{\mu_2} + r_{\nu_2} + l_2) < L_1 + L_2$, then $T_\nu^\mu(\operatorname{supp} \nu_1) \subset \operatorname{supp} \mu_1$ up to a $\nu$ negligible set.*

## 3.2 Semi-Discrete Setting

Although results in Section 3.1 provide valuable theoretical insights into OT alignment, they remain difficult to compute or verify in practical settings. In this section, we leverage the semi-discrete OT formulation to derive an equivalent condition for perfect classification under OT alignment. Building upon this, we introduce a novel quantity, OT score, that can be utilized in practice to post-check the performance of the classification from distributional alignment based DA algorithms. Also, we will show later how the following theorem inspires a way to recognize target data points classified with low confidence.

**Theorem 6.** *Suppose $\mu$ and $\nu$ are compactly supported. Then $(T_\nu^{\hat\mu})_\# \nu_1 = \hat\mu_1$ and $(T_\nu^{\hat\mu})_\# \nu_2 = \hat\mu_2$ if and only if*

$$\sup_{x \in \nu_1} \max_{z \in \hat\mu_2} \min_{y \in \hat\mu_1} \tilde{d}_{\boldsymbol{w}}(x, y) - \tilde{d}_{\boldsymbol{w}}(x, z) \leq 0 \leq \inf_{x \in \nu_2} \max_{z \in \hat\mu_2} \min_{y \in \hat\mu_1} \tilde{d}_{\boldsymbol{w}}(x, y) - \tilde{d}_{\boldsymbol{w}}(x, z),$$

*where $\tilde{d}$ is defined as in Section 2.2.2*

---

**Algorithm 1** OT score

---

1: **Input:** Source class-wise mean feature representations $\mathbf{Z}^{\mathcal{S}}$, corresponding labels $\boldsymbol{y}^{\mathcal{S}}$, source sample weights $\boldsymbol{a}$, target features $\mathbf{Z}^{\mathcal{T}}$ and corresponding predicted labels (or pseudo labels) $\hat{\boldsymbol{y}}^{\mathcal{T}}$, entropic regularization parameter $\varepsilon$, learning rate $\gamma$.

2: Initialize $\boldsymbol{w}_0 = \mathbf{0}$.

3: Compute class proportions $p_c = \frac{|\{z_i^{\mathcal{T}} \in \mathbf{Z}^{\mathcal{T}} : \hat{y}_i^{\mathcal{T}} = c\}|}{|\hat{\boldsymbol{y}}^{\mathcal{T}}|}$.

4: **for** $t = 1, 2, \ldots, max\_iter$ **do**

5:      Draw a batch of samples $\mathbf{Z}_{B_t}^{\mathcal{T}}$ from $\mathbf{Z}^{\mathcal{T}}$.

6:      Compute smoothed indicator functions of Laguerre cells $\mathbb{L}_{\boldsymbol{w}_t}(z_j^{\mathcal{S}})$ for each $z_j^{\mathcal{S}}$:

$$\chi_j^{\varepsilon}(x, \boldsymbol{w}) = \frac{e^{\frac{-\|x - z_j^{\mathcal{S}}\| + \boldsymbol{w}_t^j}{\varepsilon}}}{\sum_{\ell} e^{\frac{-\|x - z_{\ell}^{\mathcal{S}}\| + \boldsymbol{w}_t^j}{\varepsilon}}}.$$

7:      Update $\boldsymbol{w}_t$: $\boldsymbol{w}_{t+1} = \boldsymbol{w}_t - \gamma \left[ \chi_j^{\varepsilon}(x, \boldsymbol{w}_t) - a_j \right]_{j=1}^{N_{\mathcal{S}}} \in \mathbb{R}^{N_{\mathcal{S}}}$.

8: **end for**

9: **for** $(z_i^{\mathcal{T}}, \hat{\boldsymbol{y}}_i^{\mathcal{T}}) \in (\mathbf{Z}^{\mathcal{T}}, \hat{\boldsymbol{y}}^{\mathcal{T}})$ **do**

10:      Compute $g_j(x) := \max_{y \in \mathbf{X}_{\hat{\boldsymbol{y}}_i^{\mathcal{T}}}} \min_{z \in \mathbf{X}_j} \tilde{d}(x, z) - \tilde{d}(x, y)$ for each class $j$.

11:      Compute OT score of $z_i^{\mathcal{T}}$: $g(z_i^{\mathcal{T}}) = \min_j g_j(z_i^{\mathcal{T}})$

12: **end for**

---

With $\mu$ being the source measure and $\nu$ being the target measure, we define a new function $g(x) := \max_{z \in \hat{\mu}_2} \min_{y \in \hat{\mu}_1} \tilde{d}(x, y) - \tilde{d}(x, z)$. Hence, the $g$ value gap $\inf_{x \in \nu_2} g(x) - \sup_{x \in \nu_1} g(x)$ reflects the flexibility of a classification boundary induced by semi-discrete OT and a larger $g$ value gap implies better classification performance. See Figure 1 for a visual illustration.

In practice, this $g$ value gap can be used as a post-check tool once target pseudo labels have been assigned by any algorithm. We can compute the gap $\inf_{x \in \nu_1} g(x) - \sup_{x \in \nu_2} g(x)$ based on pseudo-labeled partition of the target distribution $\nu_1$ and $\nu_2$. In addition to global assessment, the individual $g(x)$ values can also serve as confidence indicators. Specifically, for target samples pseudo-labeled as class $\nu_2$, larger $g(x)$ values indicate higher classification confidence; conversely, for samples labeled as class $\nu_1$, smaller $g(x)$ values indicate higher confidence.

**Remark 3.** *Although a similar version of Theorem 6 can be derived in the discrete OT setting using analogous techniques, we choose to adopt the semi-discrete OT formulation for computing the OT score in our work, due to the following reasons:*

*(1) **Efficient incremental optimization:** Semi-discrete OT can be updated incrementally with SGD instead of being solved from scratch. As target pseudo-labels evolve, we reuse the previous solution as initialization and perform a few mini-batch SGD updates to reflect the new assignments.*

*(2) **Handling ambiguity in low-confidence filtering:** In the discrete case, there exists ambiguity in determining which points should be eliminated as low-confidence samples—whether to remove points with split weights across transport plans, or those with only small transport margins. The semi-discrete formulation mitigates such ambiguity by providing more stable and geometrically meaningful transport behavior.*

The following corollary might be helpful in some computation scenarios: it enables computing the semi-discrete OT for each component separately, thereby reducing the dimension of the dual weights.

**Corollary 7.** *Under assumptions of 6 and suppose $\boldsymbol{m}$ and $\boldsymbol{l}$ are the weight vectors associated with $T_{\nu_1}^{\hat{\mu}_1}$ and $T_{\nu_2}^{\hat{\mu}_2}$, respectively. Then $(T_{\nu}^{\hat{\mu}})_{\#} \nu_1 = \hat{\mu}_1$ and $(T_{\nu}^{\hat{\mu}})_{\#} \nu_2 = \hat{\mu}_2$ if and only if*

$$\sup_{x \in \nu_1} \max_{z \in \hat{\mu}_2} \min_{y \in \hat{\mu}_1} \tilde{d}_{\boldsymbol{m}}(x, y) - \tilde{d}_{\boldsymbol{l}}(x, z) \leq \inf_{x \in \nu_2} \max_{z \in \hat{\mu}_2} \min_{y \in \hat{\mu}_1} \tilde{d}_{\boldsymbol{m}}(x, y) - \tilde{d}_{\boldsymbol{l}}(x, z).$$

## 4 OT SCORE COMPUTATION

In this section, we extend the definition of OT score to multiclass setting and present the algorithm used for computation. Specifically, we model the source distribution in the feature space as a discrete measure and treat the target data as samples drawn from a continuous measure.

**Definition 1.** *Suppose the source data (or features)* $\mathbf{X}^{\mathcal{S}}$ *consists of c classes. For each target sample* $x$ *with pseudo label i and any class label j, we define the binary OT score as*

$$g_j(x) := \max_{y \in \mathbf{X}^{\mathcal{S}_i}} \min_{z \in \mathbf{X}^{\mathcal{S}_j}} \tilde{d}(x, z) - \tilde{d}(x, y),$$

*where* $\tilde{d}(\cdot, \cdot)$ *requires computing the semi-discrete OT. The OT score is defined as*

$$g(x) := \min_j g_j(x).$$

We summarize our OT score computation in Algorithm 1. We represent the source distribution by class-wise mean features. Accordingly, the definition of $g_j$ simplifies to $g_j(x) = \tilde{d}(x, \mu_j) - \tilde{d}(x, \mu_i)$, where $\mu_i$ and $\mu_j$ are the mean features of classes $i$ and $j$, respectively. Under this setting, we show that classification accuracy increases as samples with low OT scores are filtered out.

**Theorem 8.** *Let* $\nu_1, \nu_2$ *be the continuous probability measures with means* $m_1$ *and* $m_2$, *respectively and* $\hat{\mu}_i$ *consists of singletons* $y_i$. *Denote* $\nu := \frac{1}{2}\nu_1 + \frac{1}{2}\nu_2$ *and* $\hat{\mu} := \frac{1}{2}\hat{\mu}_1 + \frac{1}{2}\hat{\mu}_2$. *Suppose* $\nu_i(|X_i - m_i| \geq t) \leq 2\exp\left(-\frac{t^2}{2\sigma^2}\right)$ *and* $\|m_1 - y_1\| + \|m_2 - y_2\| < \|m_1 - y_2\| + \|m_2 - y_1\|$, *then*

$$P\left(T_\nu^{\hat{\mu}}(X_i) \neq Y_i | g(X_i) > g\right) \leq 2\exp\left(-\frac{\min_{i=1,2} \mathrm{dist}(m_i, \mathcal{S})^2}{2\sigma^2}\right), \text{ where}$$

*(1)* $\mathcal{S} := \left\{x : \|x - y_1\| - (w^* + g) = \|x - y_2\|\right\}$

*(2)* $d := \|y_2 - y_1\|, \qquad e := \frac{y_2 - y_1}{d}, \qquad m = \alpha e + u, \ u \perp e$ *is the orthogal decomposition of* $m$ *and denote* $\rho := \|u\|$.

*(3)*$\mathrm{dist}(m, \mathcal{S}) = \min_{r \geq 0} \sqrt{(t(r) - \alpha)^2 + (r - \rho)^2}$ *where* $t(r)$ *is defined through*

$$\sqrt{t^2 + r^2} = \sqrt{(t - d)^2 + r^2} + (w^* + g), \qquad r \geq 0.$$

## 5 APPLICATIONS AND EMPIRICAL EVALUATION

In this section, we present: (i) an Area Under the Risk–Coverage Curve (AURC) evaluation across confidence scores (Section 5.1); (ii) an SFUDA application using the OT score for training-time reweighting to improve accuracy (Section 5.2); and (iii) a label-free model-selection analysis showing that the mean OT score on the target set correlates with final accuracy (Section 5.3). Additional details and results are provided in Appendix D.

### 5.1 AURC COMPARISONS

To demonstrate the effectiveness of the proposed OT score, we compare it against several widely-used confidence estimation methods, including Maxprob, Entropy (Ent), and JMDS. The evaluation is conducted on four standard UDA benchmarks: Digits, Office-Home, ImageCLEF-DA, and VisDA-17. We compute confidence scores in the feature space extracted by the last layer of our neural network.

For evaluation, we adopt the Area Under the Risk-Coverage Curve (AURC) proposed by Geifman et al. (2018); Ding et al. (2020) and subsequently employed in Lee et al. (2022). Specifically, after obtaining the high-confidence subset $X_h^{\mathcal{T}} := \left\{x_i^{\mathcal{T}} \mid s\left(x_i^{\mathcal{T}}, \hat{y}_i^{\mathcal{T}}\right) > h\right\}$, where $h$ is a predefined confidence threshold, the risk is computed as the average empirical loss over $X_h^{\mathcal{T}}$, and the coverage corresponds to $\left|X_h^{\mathcal{T}}\right| / \left|X^{\mathcal{T}}\right|$. A lower AURC value indicates higher confidence reliability, as it implies a lower prediction risk at a given coverage level. Notably, when the $0/1$ loss is applied, a high AURC reflects a high error rate among pseudo-labels, thus indicating poor correctness and calibration of the confidence scores.

Table 1: Evaluation of confidence scores based on AURC.

| Dataset | Task | Maxprob | Ent | Cossim | JMDS | OT Score |
|---------|------|---------|-----|--------|------|----------|
| | Ar → Cl | 0.3485 | 0.3592 | 0.3013 | 0.2885 | **0.2623** |
| | Ar → Pr | 0.1697 | 0.1789 | 0.1297 | 0.1237 | **0.1208** |
| | Ar → Rw | 0.1032 | 0.1133 | 0.0897 | 0.0797 | **0.0770** |
| | Cl → Ar | 0.2686 | 0.2849 | 0.2045 | 0.2362 | **0.2020** |
| | Cl → Pr | 0.1916 | 0.2027 | **0.1182** | 0.1483 | 0.1424 |
| | Cl → Rw | 0.1703 | 0.1837 | 0.1180 | 0.1275 | **0.1179** |
| Office-Home | Pr → Ar | 0.2629 | 0.2753 | **0.1977** | 0.2123 | 0.2063 |
| | Pr → Cl | 0.3910 | 0.4052 | **0.3189** | 0.3193 | 0.3249 |
| | Pr → Rw | 0.0997 | 0.1085 | 0.0757 | 0.0786 | **0.0741** |
| | Rw → Ar | 0.1516 | 0.1621 | 0.1315 | 0.1369 | **0.1167** |
| | Rw → Cl | 0.3339 | 0.3463 | 0.2873 | 0.2664 | **0.2539** |
| | Rw → Pr | 0.0731 | 0.0796 | 0.0639 | 0.0737 | **0.0557** |
| | Avg. | 0.2137 | 0.2250 | 0.1697 | 0.1743 | **0.1628** |
| VisDA-2017 | T → V | 0.3071 | 0.3203 | 0.2780 | 0.2021 | **0.1704** |
| | C → I | 0.0515 | 0.0570 | **0.0181** | 0.0325 | 0.0252 |
| | C → P | 0.1902 | 0.1991 | 0.1579 | 0.1459 | **0.1143** |
| | I → C | 0.0099 | 0.0131 | 0.0038 | 0.0055 | **0.0036** |
| ImageCLEF-DA | I → P | 0.1198 | 0.1221 | 0.1280 | 0.1170 | **0.1000** |
| | P → C | 0.0260 | 0.0303 | **0.0062** | 0.0216 | 0.0092 |
| | P → I | 0.0347 | 0.0382 | **0.0177** | 0.0276 | 0.0186 |
| | Avg. | 0.0720 | 0.0766 | 0.0553 | 0.0583 | **0.0452** |

Maxprob and Ent use labels assigned by the pretrained source classifier while Cossim, JMDS, OT score receive pseudo labels from a Gaussian Mixture Model (GMM), following the same setup of Lee et al. (2022).

To further assess the robustness of the proposed OT score under varying pseudo-label quality, we consider another case where the pseudo labels are generated by the DSAN algorithm (Zhu et al., 2020). Under this setting, only Cossim and OT score are capable of incorporating externally generated high-quality pseudo labels. Table 5 in Appendix D shows the significant benefits of leveraging high quality pseudo labels. The OT score achieves the lowest AURC value in most adaptation tasks across the considered scenarios.

## 5.2 OT Score Reweighting

We integrate the OT score into CoWA-JMDS (Lee et al., 2022) as a per-sample weight for pseudo-labeled target instances. For each target sample $x_i$, we set

$$w_i = 2 \cdot \text{OT}(x_i) \cdot \text{JMDS}(x_i),$$

where $\text{JMDS}(x_i)$ is computed online from features during training, while $\text{OT}(x_i)$ is computed from features extracted by the *pre-adaptation* model, thereby decoupling the confidence signal from the evolving target representation. Relying solely on the same training-time features that are continually updated by pseudo-labels risks self-reinforcement (confirmation bias): incorrect pseudo-labels → representation drift → inflated "confidence" → further amplification. We mitigate this by computing the OT score from pre-adaptation features, which constrains the pseudo-label feedback loop and reduces confirmation bias. Here, the OT score is normalized to $[0, 1]$; the prefactor 2 offsets the dynamic-range compression induced by the product of two numbers in $[0, 1]$.

This integration yields higher accuracy than the original CoWA-JMDS. We evaluate on *Office-Home* (Tables 2) and *VisDA-2017* (Tables 3) in the SFUDA setting, reporting target-domain accuracy averaged over three seeds (see Appendix D).Training settings (backbone, optimizer, pseudo-labeling) follow Lee et al. (2022); the only change is the per-sample weight $w_i$.

## 5.3 Model Comparison

The OT score also serves as a *label-free* proxy for adaptation performance. This is particularly valuable when target labels are unavailable, as training accuracy on noisy pseudo-labels can be a misleading indicator (Zhang et al., 2016). At the end of adaptation training, we compute the mean

| Method | Ar→Cl | Ar→Pr | Ar→Rw | Cl→Ar | Cl→Pr | Cl→Rw | Pr→Ar | Pr→Cl | Pr→Rw | Rw→Ar | Rw→Cl | Rw→Pr | Avg |
|---|---|---|---|---|---|---|---|---|---|---|---|---|---|
| BAIT (Yang et al., 2020) | 57.4 | 77.5 | 82.4 | 68.0 | 77.2 | 75.1 | 67.1 | 55.5 | 81.9 | 73.9 | 59.5 | 84.2 | 71.6 |
| SHOT (Liang et al., 2020) | 57.1 | 78.1 | 81.5 | 68.0 | 78.2 | 78.1 | 67.4 | 54.9 | 82.2 | 73.3 | 58.8 | 84.3 | 71.8 |
| NRC (Yang et al., 2021) | 57.7 | 80.3 | 82.0 | 68.1 | 79.8 | 78.6 | 65.3 | 56.4 | 83.0 | 71.0 | 58.6 | 85.6 | 72.2 |
| ELR (Yi et al., 2023) | 58.4 | 78.7 | 81.5 | 69.2 | 79.5 | 79.3 | 66.3 | 58.0 | 82.6 | 73.4 | 59.8 | 85.1 | 72.6 |
| CPD (Zhou et al., 2024) | 59.1 | 79.0 | 82.4 | 68.5 | 79.7 | 79.5 | 67.9 | 57.9 | 82.8 | 73.8 | 61.2 | 84.6 | 73.0 |
| CoWA (Lee et al., 2022) | 56.9 | 78.4 | 81.0 | 69.1 | 80.0 | 79.9 | 67.7 | 57.2 | 82.4 | 72.8 | 60.5 | 84.5 | 72.5 |
| OTScore | 58.0 | 79.6 | 81.5 | 69.6 | 80.2 | 80.0 | 68.3 | 57.6 | 82.3 | 73.2 | 61.1 | 84.7 | 73.0 |

Table 2: Accuracy (%) on Office-Home (ResNet-50).

| Method | plane | bcycl | bus | car | horse | knife | mcycl | person | plant | sktbrd | train | truck | Avg |
|---|---|---|---|---|---|---|---|---|---|---|---|---|---|
| SFIT (Hou & Zheng, 2021) | 94.3 | 79.0 | 84.9 | 63.6 | 92.6 | 92.0 | 88.4 | 79.1 | 92.2 | 79.8 | 87.6 | 43.0 | 81.4 |
| SHOT (Liang et al., 2020) | 94.3 | 88.5 | 80.1 | 57.3 | 93.1 | 94.9 | 80.7 | 80.3 | 91.5 | 89.1 | 86.3 | 58.2 | 82.9 |
| NRC (Yang et al., 2021) | 96.8 | 91.3 | 82.4 | 62.4 | 96.2 | 95.9 | 86.1 | 80.6 | 94.8 | 94.1 | 90.4 | 59.7 | 85.9 |
| AdaCon (Chen et al., 2022) | 97.0 | 84.7 | 84.0 | 77.3 | 96.7 | 93.8 | 91.9 | 84.8 | 94.3 | 93.1 | 94.1 | 49.7 | 86.8 |
| ELR (Yi et al., 2023) | 97.3 | 89.1 | 89.8 | 79.2 | 96.9 | 97.5 | 92.2 | 82.5 | 95.8 | 94.5 | 87.3 | 34.5 | 86.4 |
| CPD (Zhou et al., 2024) | 96.7 | 88.5 | 79.6 | 69.0 | 95.9 | 96.3 | 87.3 | 83.3 | 94.4 | 92.9 | 87.0 | 58.7 | 85.8 |
| CoWA (Lee et al., 2022) | 96.2 | 89.7 | 83.9 | 73.8 | 96.4 | 97.4 | 89.3 | 86.8 | 94.6 | 92.1 | 88.7 | 53.8 | 86.9 |
| OTScore | 95.6 | 89.0 | 82.8 | 78.3 | 96.3 | 98.0 | 91.2 | 86.8 | 95.5 | 94.7 | 89.9 | 55.7 | 87.8 |

Table 3: Accuracy (%) on VisDA-2017 (ResNet-101).

OT score over the target set predictions. As shown in Fig. 2, for a fixed source domain, the mean OT score provides an ordinal proxy of post-adaptation accuracy across targets: higher mean OT corresponds to higher accuracy. Moreover, comparing *MNIST→USPS* with *FLIP-USPS→USPS* shows that a source model obtained via pixel-value inversion (FLIP-USPS) yields substantially lower SFUDA performance than using MNIST as the source as shown in Table 4.

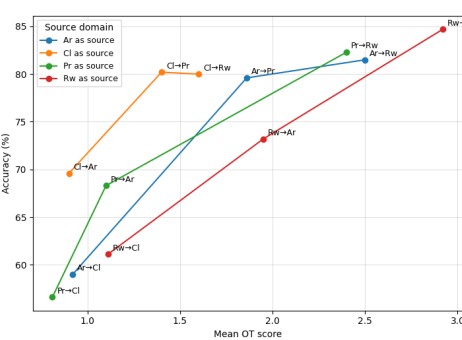

Figure 2: Mean OT Score vs. accuracy on Office-Home. Lines connect targets sharing the same source. Points denote individual target domains.

Table 4: Accuracy (%) on *USPS* with different sources.

| Source | Mean Score | Accuracy (%) |
|---|---|---|
| *MNIST* | 4.02 | 94.7 |
| *FLIP-USPS* | 0.55 | 47.8 |

## 6 CONCLUSION AND FUTURE WORK

We investigate theoretical guarantees about allowed distribution shifts in order to have a label-preserving OT. Using semi-discrete OT, we derive the OT score which considers the decision boundary induced by the OT alignment. The definition of OT score can be easily extended to other cost functions other than the standard Euclidean norm. Additionally, confidence scores are helpful for training-time sample reweighting and model comparison.

Currently, we address class imbalance in the OT-score computation by weighting the source class mean features with class proportions estimated from pseudo labels. However, when pseudo labels are unreliable, these estimates can be biased. Under the assumptions in Section E, we show that the OT objective is minimized when the source and target class proportions coincide (see Theorem 11). A natural next step is to model and propagate class-proportion uncertainty into the confidence score.

## 7 REPRODUCIBILITY STATEMENT

We have taken several steps to facilitate reproduction of our results. An anonymous code package is provided in the supplementary materials. Experimental settings—datasets, preprocessing, model architectures, hyperparameters and training schedules are summarized in Section 5 and Appendix D. Theoretical results are stated with explicit assumptions and accompanied by complete proofs in Section 3 (see also Appendix C).

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

# A    RELATED WORKS

**Theory about DA:** Several theoretical works have investigated the learnability and generalization guarantees of domain adaptation (DA). Specifically, Ben-David & Urner (2012) analyzes the DA learnability problem and sample complexity under the standard VC-dimension framework, and identifies a setting in which no algorithm can successfully solve the DA problem. In a related direction, Redko et al. (2019) provides a theoretical analysis about the existence of a hypothesis that performs well across both source and target domains, and further establishes finite-sample approximation properties of the $\lambda$ term. Le et al. (2021) alleviates the label mismatching problem by searching for a transformation $T$ that satisfies the following conditions: (1) $T_{\#}\mu_S = \mu_T$, and (2) $T$ preserves the labels.

**Confidence Scores**: Uncertainty estimation and confidence score have been prevalently employed in machine learning to improve model robustness. In particular, ordinal ranking techniques have been commonly used for selective classification (Lakshminarayanan et al. (2017); Geifman & El-Yaniv (2017); Mandelbaum & Weinshall (2017); Nair et al. (2020)), where the goal is to prioritize or filter samples based on their confidence scores in order to exclude low-confidence samples during training. Karim et al. (2023) select reliable pseudo-labels by thresholding the maximum softmax probability of the teacher's augmentation-averaged prediction. Litrico et al. (2023) reweight the classification loss by entropy, assigning higher weights to low-entropy (more confident) samples. Lee et al. (2022) propose the JMDS score to effectively identify low-confidence samples, thereby enhancing the reliability of the DA process. However, most existing confidence scores rely primarily on cluster-level information in the feature space, without explicitly modeling the geometric relationship between domains. In contrast, our proposed OT score take into account the geometry induced by the OT map, establishing a stronger connection between the source and target domains when computing confidence scores.

# B    CONFIDENCE SCORES

We provide details of the confidence scores used for comparison. Let $x_i^{\mathcal{T}}$ denote the $i$-th target sample, and let $p_{\mathcal{S}}$ represent the class probability predicted by the pretrained source model. Here, $K$ is the total number of classes, and $C_{\hat{y}_i^{\mathcal{T}}}$ denotes the center of the cluster corresponding to the predicted label $\hat{y}_i^{\mathcal{T}}$ for $x_i^{\mathcal{T}}$.

$$\text{Maxprob}\left(x_i^{\mathcal{T}}\right) = \max_c p_{\mathcal{S}}\left(x_i^{\mathcal{T}}\right)_c,$$

$$\text{Ent}\left(x_i^{\mathcal{T}}\right) = 1 + \frac{\Sigma_{c=1}^K p_{\mathcal{S}}\left(x_i^{\mathcal{T}}\right)_c \log p_{\mathcal{S}}\left(x_i^{\mathcal{T}}\right)_c}{\log K},$$

$$\text{Cossim}\left(x_i^{\mathcal{T}}\right) = \frac{1}{2}\left(1 + \frac{\left\langle x_i^{\mathcal{T}}, C_{\hat{y}_i^{\mathcal{T}}} \right\rangle}{\left\| x_i^{\mathcal{T}} \right\| \left\| C_{\hat{y}_i^{\mathcal{T}}} \right\|}\right).$$

JMDS score is computed by $\text{JMDS}\left(x_i^{\mathcal{T}}\right) = \text{LPG}\left(x_i^{\mathcal{T}}\right) \cdot \text{MPPL}\left(x_i^{\mathcal{T}}\right)$. LPG is the Log-Probability Gap computed from log data-structure-wise probability $\log p_{\text{data}}\left(x_i^{\mathcal{T}}\right)$ using GMM on the target feature space. MPPL provides high scores for samples whose GMM pseudo-label is the same based on $p_{\mathcal{S}}\left(x_i^{\mathcal{T}}\right)$ and $p_{\text{data}}\left(x_i^{\mathcal{T}}\right)$. Details about JMDS score can be found in Lee et al. (2022).

# C    PROOFS

*Proof of Theorem 4.* Due to the concentration assumptions on $\mu$ and $\nu$, we can pick sets $E_{\mu_i}$ and $E_{\nu_i}$ such that $\mu_1(E_{\mu_1}) = \mu_2(E_{\mu_2}) \geq 1 - \epsilon$. So $\frac{1}{2} + \frac{1}{2}\epsilon \geq \mu(E_{\mu_i}) \geq \frac{1}{2} - \frac{1}{2}\epsilon$. The same holds for $\nu(E_{\nu_i})$.

Consider $F_i = (T_\nu^\mu)^{-1}(E_{\mu_i})$, we have $\nu(F_i) = \mu(E_{\mu_i}) \geq \frac{1}{2} - \frac{1}{2}\epsilon$ as well. Let $F = F_1 \cup F_2$. So $E_{\nu_i} \cap F$ is a bounded set with

$$\frac{1}{2} + \frac{1}{2}\epsilon \geq \nu(E_{\nu_i} \cap F) = \nu(E_{\nu_i}) - \nu(E_{\nu_i} \cap F^c) \tag{1}$$

$$\geq \frac{1}{2} - \frac{1}{2}\epsilon - \nu(F^c) \tag{2}$$

$$\geq \frac{1}{2} - \frac{1}{2}\epsilon - \epsilon = \frac{1}{2} - \frac{3}{2}\epsilon \tag{3}$$

Without loss of generality, we assume $\nu(E_{\nu_1} \cap F) \geq \nu(E_{\nu_2} \cap F)$. Since $\nu \ll \mathcal{L}$, we can pick $R > 0$ such that $\nu(E_{\nu_1} \cap F \cap B_R) = \nu(E_{\nu_2} \cap F)$.

Now consider the optimal transport map $T_\nu^\mu$ restricted on $(E_{\nu_1} \cap F \cap B_R) \cup (E_{\nu_2} \cap F)$. By (Villani et al., 2009, Theorem 4.6), this restricted map is an optimal transport map between the marginal measures.

Since $\mu\big(T_\nu^\mu(E_{\nu_1} \cap F \cap B_R) \cup T_\nu^\mu(E_{\nu_2} \cap F)\big) = \nu\big((E_{\nu_1} \cap F \cap B_R) \cup (E_{\nu_2} \cap F)\big) \geq 1 - 3\epsilon$, we get an estimate $\mu\big((T_\nu^\mu(E_{\nu_1} \cap F \cap B_R) \cup T_\nu^\mu(E_{\nu_2} \cap F)) \cap E_{\mu_i}\big) \geq (1 - 3\epsilon) - (\frac{1}{2} + \frac{1}{2}\epsilon) = \frac{1}{2} - \frac{7}{2}\epsilon$. Therefore, we can use Lemma 9 to conclude that with probability greater than $1 - 7\epsilon$, target samples will be correctly classified after optimal transportation.

$\square$

**Lemma 9.** *Suppose we have probability measures $\mu_i$ and $\nu_i$ with bounded support. Also assume that* $\operatorname{supp}\mu_1$ *and* $\operatorname{supp}\mu_2$ *are disjoint,* $\operatorname{supp}\nu_1$ *and* $\operatorname{supp}\nu_2$ *are disjoint. Let $r_{\mu_i}$ denote the diameter of the support of $\mu_i$ and set $l_i = d(\operatorname{supp}\mu_i, \operatorname{supp}\nu_i)$, $L_1 = d(\operatorname{supp}\mu_1, \operatorname{supp}\nu_2)$, $L_2 = d(\operatorname{supp}\mu_2, \operatorname{supp}\nu_1)$. Suppose $\mu := \frac{1}{2}\mu_1 + \frac{1}{2}\mu_2$, $\nu := p\nu_1 + (1-p)\nu_2$ for some $p \in (0, \frac{1}{2}]$. If $(r_{\mu_1} + r_{\nu_1} + l_1) + (r_{\mu_2} + r_{\nu_2} + l_2) < L_1 + L_2$, then $T_\nu^\mu(\operatorname{supp}\nu_1) \subset \operatorname{supp}\mu_1$ up to a negligible set.*

*Proof of Lemma 9.* Suppose there exists a set $A \subset \operatorname{supp}\nu_1$ with $\nu(A) = \delta > 0$ and $T_\nu^\mu(A) \subset \operatorname{supp}\mu_2$. Then there must be a set $B \subset \operatorname{supp}\nu_2$ with $\nu(B) \geq \delta + 1 - p - \frac{1}{2} = \frac{1}{2} + \delta - p$ and $T_\nu^\mu(B) \subset \operatorname{supp}\mu_1$. Since $\nu_i \ll \mathcal{L}$, we can pick $B' \subset B$ such that $\nu(B') = \delta$. Then for any measurable $\tilde{T}$ such that $\tilde{T}(A) = T_\nu^\mu(B')$ and $\tilde{T}(B') = T_\nu^\mu(A)$,

$$\int_{A \cup B'} \|\tilde{T}(x) - x\| \, dx \leq \delta(r_{\mu_1} + r_{\nu_1} + l_1) + \delta(r_{\mu_2} + r_{\nu_2} + l_2) < \delta(L_1 + L_2) \leq \int_{A \cup B'} \|T_\nu^\mu(x) - x\| \, dx,$$

which contradicts the optimality of $T_\nu^\mu$.

$\square$

*Proof of Theorem 6.* Let $\boldsymbol{w}$ be any weight vector associated with $T_\nu^{\hat{\mu}}$. We start with the observation that $(T_\nu^{\hat{\mu}})_\# \bar{\nu}_1 = \hat{\mu}_1$ and $(T_\nu^{\hat{\mu}})_\# \bar{\nu}_2 = \hat{\mu}_2$ is equivalent to the following two conditions:

(1) For $\forall x \in \bar{\nu}_1$, $\tilde{d}_{\boldsymbol{w}}(x, \hat{\mu}_1) \leq \tilde{d}_{\boldsymbol{w}}(x, \hat{\mu}_2)$.

(2) And for $\forall x \in \bar{\nu}_2$, $\tilde{d}_{\boldsymbol{w}}(x, \hat{\mu}_2) \leq \tilde{d}_{\boldsymbol{w}}(x, \hat{\mu}_1)$.

(1) requires any point from $\bar{\nu}_1$ to be assigned to some point in $\hat{\mu}_1$ and (2) requires any point from $\bar{\nu}_2$ to be assigned to some point in $\hat{\mu}_2$, i.e.

$$\sup_{x \in \bar{\nu}_1} \tilde{d}_{\boldsymbol{w}}(x, \hat{\mu}_1) - \tilde{d}_{\boldsymbol{w}}(x, \hat{\mu}_2) \leq 0 \leq \inf_{x \in \bar{\nu}_2} \tilde{d}_{\boldsymbol{w}}(x, \hat{\mu}_1) - \tilde{d}_{\boldsymbol{w}}(x, \hat{\mu}_2). \tag{4}$$

We rewrite 4 by unwarpping the definition of $\tilde{d}$ to get

$$\sup_{x \in \bar{\nu}_1} \left( \min_{y \in \hat{\mu}_1} \tilde{d}_{\boldsymbol{w}}(x, y) \right) - \left( \min_{z \in \hat{\mu}_2} \tilde{d}_{\boldsymbol{w}}(x, z) \right) \leq 0 \leq \inf_{x \in \bar{\nu}_2} \left( \min_{y \in \hat{\mu}_1} \tilde{d}_{\boldsymbol{w}}(x, y) \right) - \left( \min_{z \in \hat{\mu}_2} \tilde{d}_{\boldsymbol{w}}(x, z) \right), \tag{5}$$

i.e.

$$\sup_{x \in \bar{\nu}_1} \max_{z \in \hat{\mu}_2} \min_{y \in \hat{\mu}_1} \tilde{d}_{\boldsymbol{w}}(x, y) - \tilde{d}_{\boldsymbol{w}}(x, z) \leq 0 \leq \inf_{x \in \bar{\nu}_2} \max_{z \in \hat{\mu}_2} \min_{y \in \hat{\mu}_1} \tilde{d}_{\boldsymbol{w}}(x, y) - \tilde{d}_{\boldsymbol{w}}(x, z). \tag{6}$$

$\square$

*Proof of Corollary 7.* Observe that $\boldsymbol{w}_1 = \boldsymbol{m} + C$ and $\boldsymbol{w}_2 = \boldsymbol{l} + D$ are also weight vectors for $T_{\bar{\nu}_1}^{\hat{\mu}_1}$ and $T_{\bar{\nu}_2}^{\hat{\mu}_2}$ for any constants $C$ and $D$.

Moreover,

$$\sup_{x \in \bar{\nu}_1} \max_{z \in \hat{\mu}_2} \min_{y \in \hat{\mu}_1} \tilde{d}_{\boldsymbol{w}_1}(x,y) - \tilde{d}_{\boldsymbol{w}_2}(x,z) \leq \inf_{x \in \bar{\nu}_2} \max_{z \in \hat{\mu}_2} \min_{y \in \hat{\mu}_1} \tilde{d}_{\boldsymbol{w}_1}(x,y) - \tilde{d}_{\boldsymbol{w}_2}(x,z),$$

which is the same as

$$C - D + \sup_{x \in \bar{\nu}_1} \max_{z \in \hat{\mu}_2} \min_{y \in \hat{\mu}_1} \tilde{d}_{\boldsymbol{m}}(x,y) - \tilde{d}_{\boldsymbol{l}}(x,z) \leq C - D + \inf_{x \in \bar{\nu}_2} \max_{z \in \hat{\mu}_2} \min_{y \in \hat{\mu}_1} \tilde{d}_{\boldsymbol{m}}(x,y) - \tilde{d}_{\boldsymbol{l}}(x,z). \quad (7)$$

Choosing the difference $C - D$ allows us to conclude $(T_{\nu}^{\hat{\mu}})_{\#}\bar{\nu}_1 = \hat{\mu}_1$ and $(T_{\nu}^{\hat{\mu}})_{\#}\bar{\nu}_2 = \hat{\mu}_2$ by setting $\boldsymbol{w} = [\boldsymbol{w}_1, \boldsymbol{w}_2]$.

Conversely, let $\boldsymbol{w}$ be the corresponding weight vector of $T_{\nu}^{\hat{\mu}}$ and assume $(T_{\nu}^{\hat{\mu}})_{\#}\bar{\nu}_1 = \hat{\mu}_1$, $(T_{\nu}^{\hat{\mu}})_{\#}\bar{\nu}_2 = \hat{\mu}_2$. Then $\boldsymbol{w}_1$ (or $\boldsymbol{w}_2$) differs from $\boldsymbol{m}$ (or $\boldsymbol{l}$) by some constant $C$ (or $D$) (Geiß et al., 2013, Theorem 2). By Theorem 6,

$$\sup_{x \in \bar{\nu}_1} \max_{z \in \hat{\mu}_2} \min_{y \in \hat{\mu}_1} \tilde{d}_{\boldsymbol{w}_1}(x,y) - \tilde{d}_{\boldsymbol{w}_2}(x,z) \leq 0 \leq \inf_{x \in \bar{\nu}_2} \max_{z \in \hat{\mu}_2} \min_{y \in \hat{\mu}_1} \tilde{d}_{\boldsymbol{w}_1}(x,y) - \tilde{d}_{\boldsymbol{w}_2}(x,z),$$

which implies

$$C - D + \sup_{x \in \bar{\nu}_1} \max_{z \in \hat{\mu}_2} \min_{y \in \hat{\mu}_1} \tilde{d}_{\boldsymbol{m}}(x,y) - \tilde{d}_{\boldsymbol{l}}(x,z) \leq 0 \leq C - D + \inf_{x \in \bar{\nu}_2} \max_{z \in \hat{\mu}_2} \min_{y \in \hat{\mu}_1} \tilde{d}_{\boldsymbol{m}}(x,y) - \tilde{d}_{\boldsymbol{l}}(x,z),$$

i.e.

$$\sup_{x \in \bar{\nu}_1} \max_{z \in \hat{\mu}_2} \min_{y \in \hat{\mu}_1} \tilde{d}_{\boldsymbol{m}}(x,y) - \tilde{d}_{\boldsymbol{l}}(x,z) \leq D - C \leq \inf_{x \in \bar{\nu}_2} \max_{z \in \hat{\mu}_2} \min_{y \in \hat{\mu}_1} \tilde{d}_{\boldsymbol{m}}(x,y) - \tilde{d}_{\boldsymbol{l}}(x,z).$$

$\square$

This proposition shows how the classification accuracy improves with samples conditioned on high confidence scores $\Delta w$.

**Theorem 10.** *Let $\nu_1, \nu_2$ be the continuous probability measures with means $m_1$ and $m_2$, respectively and $\hat{\mu}_i$ consists of singletons $y_i$. Denote $\nu := \frac{1}{2}\nu_1 + \frac{1}{2}\nu_2$ and $\hat{\mu} := \frac{1}{2}\hat{\mu}_1 + \frac{1}{2}\hat{\mu}_2$. Suppose $\nu_i(|X_i - m_i| \geq t) \leq 2\exp\left(-\frac{t^2}{2\sigma^2}\right)$ and $\|m_1 - y_1\| + \|m_2 - y_2\| < \|m_1 - y_2\| + \|m_2 - y_1\|$, then*

$$P\left(T_{\nu}^{\hat{\mu}}(X_i) \neq Y_i | g(X_i) > \Delta w\right) \leq 2\exp\left(-\frac{\min_{i=1,2} \mathrm{dist}(m_i, \mathcal{S})^2}{2\sigma^2}\right), \text{ where}$$

*(1)* $\mathcal{S} := \left\{x : \|x - y_1\| - (w^* + \Delta w) = \|x - y_2\|\right\}$

*(2)* $d := \|y_2 - y_1\|, \qquad e := \frac{y_2 - y_1}{d}, \qquad m = \alpha e + u, \ u \perp e$ *is the orthogal decomposition of $m$ and denote $\rho := \|u\|$.*

*(3)* $\mathrm{dist}(m, \mathcal{S}) = \min_{r \geq 0} \sqrt{\left(t(r) - \alpha\right)^2 + \left(r - \rho\right)^2}$ *where $t(r)$ is defined through*

$$\sqrt{t^2 + r^2} = \sqrt{(t-d)^2 + r^2} + (w^* + \Delta w), \qquad r \geq 0.$$

*Proof.* Let $w^*$ be the dual weight corresponding to $T_{\nu}^{\hat{\mu}}$ and let $w := w + \Delta w$. Denote $\mathbb{L}_{\boldsymbol{w}}(y_1) := \left\{x : \|x - y_1\| - w \leq \|x - y_2\|\right\}$ and similarly for $\mathbb{L}_{\boldsymbol{w}}(y_2)$.

Define $\mathcal{S} := \left\{x : \|x - y_1\| - w = \|x - y_2\|\right\}$. Without loss of generality, we assume $y_1 = \boldsymbol{0}$. For an arbitrary point $m \in \mathbb{R}^n$, write the orthogonal decomposition

$$d := \|y_2\|, \qquad e := \frac{y_2}{d}, \qquad m = \alpha e + u, \ u \perp e, \ \rho := \|u\|.$$

For every $x$ write

$$x = t e + v, \qquad t \in \mathbb{R}, \ v \perp e, \ r := \|v\|.$$

Under this decomposition

$$\|x\| = \sqrt{t^2 + r^2}, \qquad \|x - y_2\| = \sqrt{(t-d)^2 + r^2}.$$

Hence $x \in \mathcal{S}$ iff

$$\sqrt{t^2 + r^2} = \sqrt{(t-d)^2 + r^2} + w, \qquad r \geq 0. \tag{8}$$

Since for any fixed $r$, $\sqrt{t^2 + r^2} - \sqrt{(t-d)^2 + r^2}$ is strictly increasing, solution to equation 8 is unique and we denote it by $t(r)$.

The squared distance between $x = t\,e + v$ and $m$ is

$$\|x - m\|^2 = (t - \alpha)^2 + \|v - u\|^2 = (t - \alpha)^2 + r^2 + \rho^2 - 2r\rho\cos\theta,$$

where $\theta$ is the angle between $v$ and $u$. For fixed $(t, r)$ this expression is minimized when $\theta = 0$, i.e. $v$ is chosen to be colinear with $u$. Without loss of generality set $v = (r/\rho)\,u$ when $\rho \neq 0$.

The minimal squared distance at any given $(t, r)$ is therefore $(t - \alpha)^2 + (r - \rho)^2$. Since $t = t(r)$ is uniquely determined by $r$, the distance optimization reduces to

$$\mathrm{dist}(m, \mathcal{S}) = \min_{r \geq 0} \sqrt{\left(t(r) - \alpha\right)^2 + \left(r - \rho\right)^2}.$$

By a direct derivative analysis, the minimizer for $dist(m, \mathcal{S})$ is unique.

Therefore, take $m = m_1$, we have $\nu_1(\mathbb{L}_{\boldsymbol{w}}(y_1)) \geq 1 - 2\exp\left(-\frac{\mathrm{dist}(m_1, \mathcal{S})^2}{2\sigma^2}\right)$.

$\square$

# D    EXPERIMENT DETAILS

## D.1    SYNTHETIC DATA

In this section, we use synthetic data to validate and illustrate our theoretical findings. Specifically, we consider a 2D scenario where data points are sampled from circular regions. The source domain consists of class-separated samples drawn from disjoint circles, whereas the target domain includes clusters with partial overlap. The distribution of the generated data is visualized in Figure 3(a). We compute the max-min values as described in Theorem 6 and present the results in Figure 3(b). As shown in Figure 3(c), many of the generated pseudo labels within the overlapping region are misclassified. However, after removing low-confidence predictions, the remaining samples are almost entirely classified correctly, as illustrated in Figure 3(d). Notably, the separation between the two clusters becomes significantly more obvious after this filtering step.

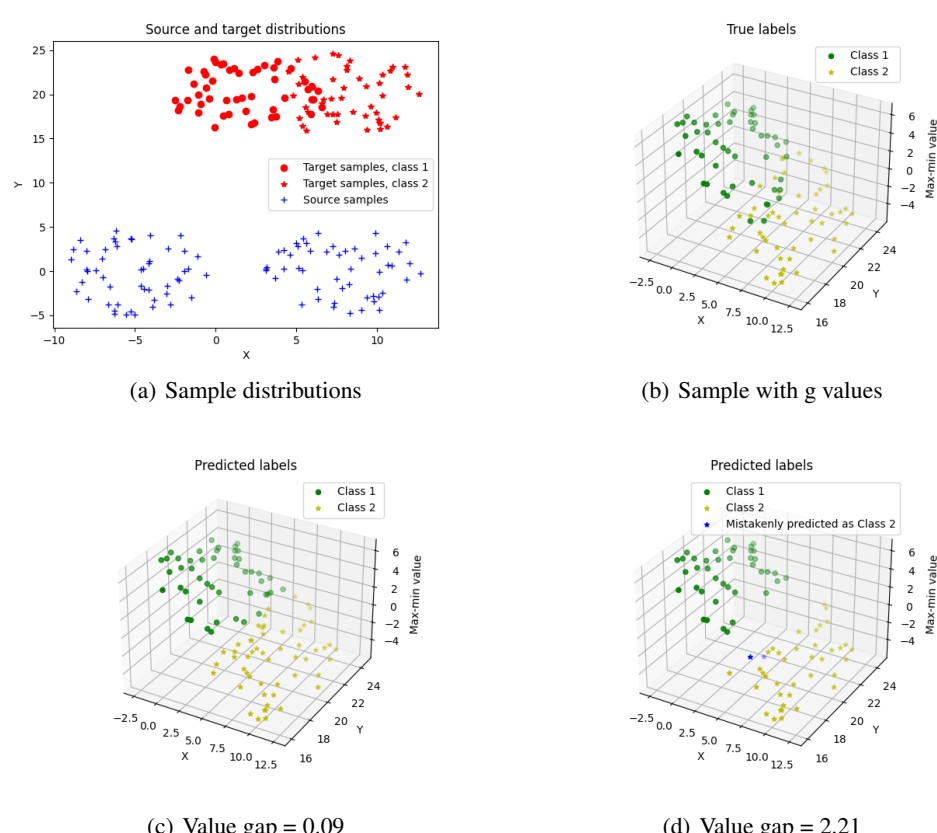

(a) Sample distributions

(b) Sample with g values

(c) Value gap = 0.09

(d) Value gap = 2.21

Figure 3: OT score performance on overlapping distributions.

## D.2 REAL-WORLD DATASETS

To ensure a fair comparison, we following the training setting of Lee et al. (2022). In our main experiments, we compare OT score with other confidence scores including Maxprob, Ent, Cossim, and JMDS. The details for other confidence scores are presented in Appendix B. We compare the performance of confidence scores on four standard UDA benchmarks: ImageCLEFDA, Office-Home, and VisDA-2017. All code can be efficiently executed on a single NVIDIA RTX 4070 GPU without requiring specialized hardware. For ImageCLEFDA, Office-Home datasets, we use ResNet-50 backbone pretrained on the ImageNet as a base network. The source model is trained for 50 epochs. For VisDA-2017, we use ResNet-101 for GMM pseudo labeling and ResNet-50 for DSAN pseudo labeling. The source model is obtained by finetuning a pretrained network on the source domain for 10 epochs. We use SGD optimizer with the momentum term set to be 0.9. We set lr=1e-4 for the base network and lr=1e-3 for the classifier layer. For digit recognition tasks, we use the ResNet-18 network as the base model. The network is initialized with random weights. We finetune this network on source domains using lr=1e-4, epochs=50, momentum=0.9, decay=1e-4. For OT score computation, we fix the entropic regularization parameter $\varepsilon$ to be 0.0001.

Pseudo-label generation via DSAN: To obtain pseudo labels, we need to further train the neural network using the DSAN algorithm with the following settings: number of training epochs = 20, `transfer_loss_weight` = 0.5, `transfer_loss` = LMMD, learning rate = 0.01, weight decay = $5 \times 10^{-4}$, momentum = 0.9. `lr_scheduler` is enabled with `lr_gamma` = 0.0003, `lr_decay` = 0.75. comparison for DSAN generated pseudo labels are provided in Table 5.

We report mean ± standard deviation over three independent runs (random seeds) in Table 6 for Office-Home and Table 7 for VisDA-2017.

| Method | Ar→Cl | Ar→Pr | Ar→Rw | Cl→Ar | Cl→Pr | Cl→Rw | |
|--------|-------|-------|-------|-------|-------|-------|---|
| OTScore | $58.0 \pm 0.6$ | $79.6 \pm 0.1$ | $81.5 \pm 0.1$ | $69.6 \pm 0.4$ | $80.2 \pm 0.8$ | $80.0 \pm 0.2$ | |

| Method | Pr→Ar | Pr→Cl | Pr→Rw | Rw→Ar | Rw→Cl | Rw→Pr | Avg |
|--------|-------|-------|-------|-------|-------|-------|-----|
| OTScore | $68.3 \pm 0.5$ | $57.6 \pm 0.7$ | $82.3 \pm 0.4$ | $73.2 \pm 0.1$ | $61.1 \pm 0.9$ | $84.7 \pm 0.4$ | $73.0 \pm 0.1$ |

Table 6: Accuracy (%) on Office-Home (ResNet-50).

| Dataset | Mean Accuracy | Classwise Mean Accuracy |
|---------|---------------|-------------------------|
| VisDA-2017 | $85.0 \pm 0.3$ | $87.8 \pm 0.1$ |

Table 7: Accuracy (%) on VisDA-2017.

Table 5: Evaluation of confidence scores based on AURC (DSAN).

| Dataset | Task | Maxprob | Ent | Cossim | JMDS | OT Score |
|---------|------|---------|-----|--------|------|----------|
| ImageCLEF-DA | $C \to I$ | 0.0301 | 0.0318 | 0.0506 | 0.0258 | **0.0240** |
| | $C \to P$ | 0.2024 | 0.2040 | 0.1913 | 0.1391 | **0.1331** |
| | $I \to C$ | 0.0090 | 0.0109 | **0.0084** | 0.0105 | 0.0090 |
| | $I \to P$ | 0.1135 | 0.1120 | 0.1607 | 0.1223 | **0.1119** |
| | $P \to C$ | 0.0102 | 0.0121 | **0.0075** | 0.0096 | 0.0097 |
| | $P \to I$ | 0.0136 | 0.0150 | 0.0186 | 0.0140 | **0.0135** |
| | **Avg.** | 0.0631 | 0.0643 | 0.0729 | 0.536 | 0.0502 |
| Office-Home | $Ar \to Cl$ | 0.4306 | 0.4284 | 0.4170 | 0.4515 | **0.3403** |
| | $Ar \to Pr$ | 0.2745 | 0.2738 | 0.2512 | 0.2849 | **0.2133** |
| | $Ar \to Rw$ | 0.1469 | 0.1493 | 0.1521 | 0.1860 | **0.1157** |
| | $Cl \to Ar$ | 0.2600 | 0.2631 | 0.2340 | 0.3228 | **0.2097** |
| | $Cl \to Pr$ | 0.1757 | 0.1777 | 0.1612 | 0.2225 | **0.1503** |
| | $Cl \to Rw$ | 0.1834 | 0.1848 | 0.1865 | 0.2246 | **0.1493** |
| | $Pr \to Ar$ | 0.2371 | 0.2381 | 0.2245 | 0.2776 | **0.1984** |
| | $Pr \to Cl$ | 0.3139 | 0.3105 | 0.3149 | 0.3302 | **0.2711** |
| | $Pr \to Rw$ | 0.0974 | 0.0992 | 0.1037 | 0.1250 | **0.0817** |
| | $Rw \to Ar$ | 0.1301 | 0.1318 | 0.1268 | 0.1751 | **0.1023** |
| | $Rw \to Cl$ | 0.2581 | 0.2555 | 0.2641 | 0.2718 | **0.2112** |
| | $Rw \to Pr$ | 0.0681 | 0.0684 | 0.0628 | 0.1026 | **0.0561** |
| | **Avg.** | 0.2146 | 0.2150 | 0.2082 | 0.2478 | **0.1749** |
| VisDA-2017 | $T \to V$ | 0.2301 | 0.2290 | 0.2289 | 0.2296 | **0.1799** |

# E  UNBALANCED CLASSES

**Theorem 11.** *With the same notations of 9, suppose $\mu = p^*\mu_1 + (1 - p^*)\mu_2$ for some $p^* \in (0, 1)$. If $L_i \geq l_i + r_{\nu_1} + r_{\nu_2} + r_{\mu_1} + r_{\mu_2}$ then $\arg\min_{p \in [0,1]} W_1(\mu, \nu) = p^*$, where $\nu := p\nu_1 + (1 - p)\nu_2$ for some $p \in (0, 1)$.*

*Proof.* W.L.O.G we assume $p^* = \frac{1}{2}$. Let $T$ denote an OT map between $\frac{1}{2}\nu_1 + \frac{1}{2}\nu_2$ and $\frac{1}{2}\mu_1 + \frac{1}{2}\mu_2$. Suppose $\nu = (\frac{1}{2} + \delta)\nu_1 + (\frac{1}{2} - \delta)\nu_2$. Let $F_1$ be the set such that $F_1 \subset \operatorname{supp} \nu_1$ and $\nu_1(F_1) = \frac{2\delta}{1+2\delta}$ so that $(\frac{1}{2} + \delta)\nu_1(F_1^C) = \frac{1}{2}$. Let $F_2 \subset \operatorname{supp} \mu_2$ be defined as $F_2 := T_\nu^\mu(F_1)$. This can be done due

to Lemma 9. Given Lemma 9, it suffices to show the following inequality:

$$\int_{F_1} \|T_\nu^\mu(x) - x\| d((\tfrac{1}{2} + \delta)\nu_1) + \int_{F_1^C} \|T_\nu^\mu(x) - x\| d((\tfrac{1}{2} + \delta)\nu_1) + \int_{\text{supp}\,\nu_2} \|T_\nu^\mu(x) - x\| d((\tfrac{1}{2} - \delta)\nu_2)$$

$$\geq W_1(\tfrac{1}{2}\nu_1, \tfrac{1}{2}\mu_1) + W_1(\tfrac{1}{2}\nu_2, \tfrac{1}{2}\mu_2).$$

Denote $\bar{\mu}_2 := (T_\nu^\mu)_{\#}((\tfrac{1}{2} - \delta)\nu_2)$. We can decompose $W_1(\tfrac{1}{2}\nu_2, \tfrac{1}{2}\mu_2) = a + b$ where $a$ corresponds to the cost on the source probability mass that forms $\bar{\mu}_2$ and $b$ corresponds to the cost on the rest of source probability mass. We denote the source marginal corresponding to $a$ as $\tfrac{1}{2}\tilde{\nu}_2$. Then it remains to show

$$\int_{F_1} \|T_\nu^\mu(x) - x\| d((\tfrac{1}{2} + \delta)\nu_1) - b$$

$$\geq W_1(\tfrac{1}{2}\nu_1, \tfrac{1}{2}\mu_1) - \int_{F_1^C} \|T_\nu^\mu(x) - x\| d((\tfrac{1}{2} + \delta)\nu_1)$$

$$+ a - \int_{\text{supp}\,\nu_2} \|T_\nu^\mu(x) - x\| d((\tfrac{1}{2} - \delta)\nu_2)$$

Note $\int_{F_1^C} \|T_\nu^\mu(x) - x\| d((\tfrac{1}{2} + \delta)\nu_1)$ achieves the optimal transport between $(\tfrac{1}{2} + \delta)\nu_1$ restricted on $F_1^C$ and $\tfrac{1}{2}\mu_1$. Also, $\int_{\text{supp}\,\nu_2} \|T_\nu^\mu(x) - x\| d((\tfrac{1}{2} - \delta)\nu_2)$ achieves the optimal transport between $(\tfrac{1}{2} - \delta)\nu_2$ and $\bar{\mu}_2$. By triangle inequality properties of $W_1$ distance, it suffices to show

$$LHS \geq W_1(\tfrac{1}{2}\nu_1, (\tfrac{1}{2} + \delta)\nu_1|_{F_1^C}) + W_1(\tfrac{1}{2}\tilde{\nu}_2, (\tfrac{1}{2} - \delta)\nu_2).$$

Since

$$RHS \leq \delta r_{\nu_1} + \delta r_{\nu_2} \leq LHS,$$

the optimality is proved. $\qquad\square$

We verify Theorem 11 with synthetic data generated within two circular clusters. We compute (discrete) OT plans under unbalanced cluster settings; see Figure 4 and Figure 5. In this experiment, we generate two equally sized clusters for the target samples, while the corresponding source clusters are assigned proportions of $0.2$ and $0.8$, respectively. As shown in the results, the optimal transport cost is minimized when the reweighting factor is correctly set to $p = 0.2$. This observation supports our claim that optimizing the reweighting factor can effectively mitigate class imbalance in optimal transport–based domain adaptation. However, this finding has not yet been validated on real-world datasets, where the underlying distributions are significantly more complex. We leave this investigation for future work.

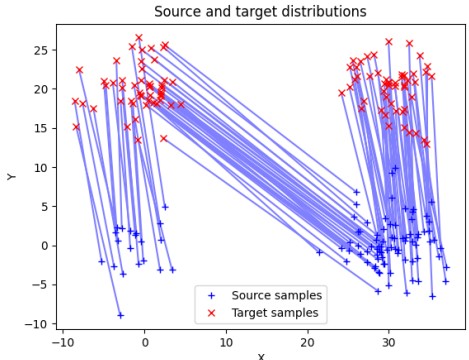

Figure 4: Unbalanced clusters with p=0.5

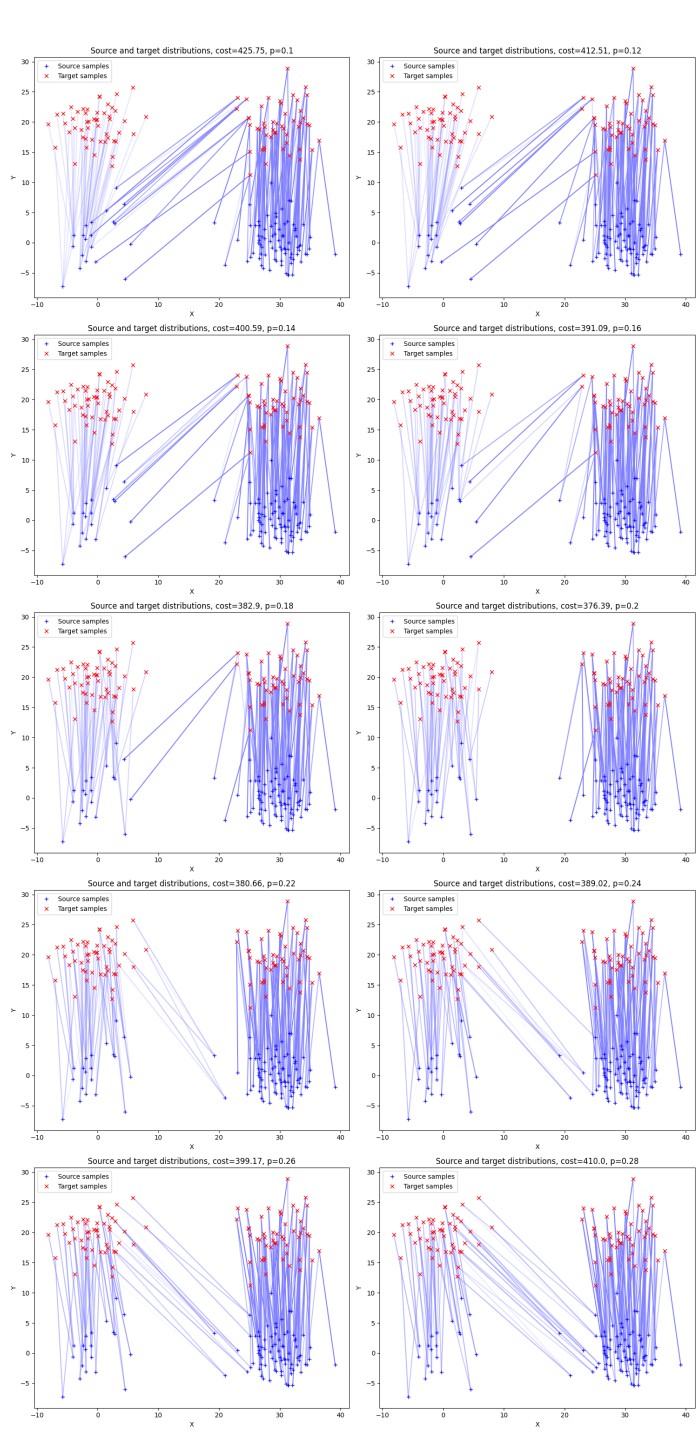

Figure 5: Unbalanced clusters

## F   The Use of Large Language Models

We used a large language model (ChatGPT) only to aid with grammar, wording, and stylistic polishing of text. All ideas, results, and claims are our own; we manually verified factual statements and citations. Only non-sensitive draft text was provided to the tool, and all outputs were reviewed and edited by the authors. Any remaining errors are our responsibility.

