# OpenReview forum: "OT Score: An OT based Confidence Score for Source Free Unsupervised Domain Adaptation"
_ICLR.cc/2026/Conference — ICLR 2026 Conference Withdrawn Submission_

### Official Review · Reviewer_gSf2 · 2025-11-01

**Soundness:** 2
**Presentation:** 1
**Contribution:** 2
**Rating:** 2
**Confidence:** 4

**Summary:**

This paper introduces an optimal transport-based confidence score (OT Score) for Source-Free Unsupervised Domain Adaptation (SFUDA).

**Strengths:**

1. The idea of quantifying pseudo-label reliability using OT geometry provides a potentially principled view of uncertainty estimation in SFUDA.

2. The authors attempt to connect OT distance and pseudo-label confidence through several lemmas and theorems, giving some formal interpretation of the metric.

**Weaknesses:**

1. **Presentation and mathematical rigor.**
    The paper’s theoretical exposition is not rigorous and suffers from missing definitions and inconsistent notation.

    - In Section 2.2.1, several key mathematical objects are undefined, including the *transport map* and the specific optimization problem whose minimizer defines the *optimal transport plan*.

    - In Theorem 2, the notation $\mathrm{Lip}_b(\mathbb{R}^d)$ is not defined.

    - Theorem 3 refers to the set $O(\mu,\nu)$ without an explicit expression; it is also unclear how the use of the Euclidean cost $\|x - y\|_2$ relates to  “$p = 1$” in Line 173.

    - Theorem 4 lacks precision: (1) the index $i \in \{1,2\}$ should be explicitly stated; (2) it is unclear whether $\epsilon \in (0,1)$ is an arbitrary constant or a fixed parameter; (3) the definitions of the set *diameter* and the set distance $d(\cdot,\cdot)$ are missing; and (4) the phrase “target samples … after the optimal transport” is vague and should be accompanied by a formal expression.

    - In Theorem 6: (1) the notation $T\^{\hat{\mu}}\_{\nu}$ seems to denote the optimal transport mapping from $\nu$ to $\hat{\mu}$ under cost $\|x-y\|^p$, but this is never made explicit; (2) It seems that the vector $\mathbf{w}$ is the optimal in $\mathbb{R}^m$ corresponding to $T^{\hat{\mu}}_{\nu}$, but it is not stated; (2) The measures used in the theorem are also not defined or described; (3) Furthermore, expressions like $x \in \nu_1$ are mathematically ambiguous, since $x$ is a point and $\nu_1$ is a probability measure; (4) The binary class correspondence of subscripts “1” and “2” should also be explicitly clarified.

    - Theorem 8 is even less clear: the function $g$ is used both as a function and a scalar constant; $P(\cdot)$ lacks a defined probability space; and symbols such as $w^\*$ and $\alpha$ are undefined.

2. **Algorithmic ambiguity.**

    - Despite the title claiming a *source-free* setting, Algorithm 1 lists “source data” as part of its input, contradicting the stated setup.

    - Step 6 involves the *smoothed indicator functions of Laguerre cells*, but no justification or reference is given. This step should explicitly cite *Peyré et al. (2019)*.

3. **Theoretical motivation and assumptions.**

    - Section 2.2.1 restates results from *Santambrogio (2015)* (Theorem 3) but never uses them in subsequent derivations. The purpose of including these results is unclear.

    - Theorem 4 assumes “disjoint bounded sets,” which is extremely restrictive and unrealistic for practical domain adaptation problems, where source and target supports typically overlap.

**Questions:**

See above.

---

### Official Review · Reviewer_m9qp · 2025-11-07

**Soundness:** 1
**Presentation:** 1
**Contribution:** 2
**Rating:** 2
**Confidence:** 3

**Summary:**

The paper aims to address the computational and theoretical limitations of current distributional
alignment methods for source-free unsupervised domain adaptation (SFUDA), with the
focus on estimating classification performance and confidence in
the absence of target labels. The authors intend to answer the questions: i) under what conditions on the domain shift can the target distribution be realigned to the source while preserving class labels; and (ii) given only potentially noisy target pseudo-labels, is there a theoretically justified and computable metric to quantify violations of these conditions?
They
propose an Optimal Transport (OT) based confidence score, which leverages a semi-discrete OT formulation to measure confidence in pseudo-labels.
While the topic is interesting, the paper has significant shortcomings in  theoretical rigor
 and generality.

**Strengths:**

The manuscript

-  considers an important problem in domain adaptation, where confidence estimation is critical;

-  presents a necessary condition on the target data distribution under which correct classification
is expected after applying optimal transport; and

- attempts to provide a theoretically motivated approach using semi-discrete OT.

**Weaknesses:**

- The development relies heavily on restrictive assumptions (e.g., bounded, finite supports and
Neural Collapse conditions).

- Notation is not used or defined consistently, and essential details are missing. The claimed
guarantees appear largely qualitative and are difficult to assess. Several passages seem isolated
and lack clear logical integration.

- The cited results (e.g., Theorems 2–3) are not well presented or clearly linked to the proposed
work. In Theorem 1, some quantities are undefined, making the motivation for the stated
challenges vague.

- The method’s theoretical justification assumes access to reliable pseudo-labels, creating a
circular dependency between the confidence score and classifier accuracy.

-  Experimental results do not appear to be completely reported.

- Inconsistent notation, missing hyperparameter details, and grammatical errors reduce readability.

**Questions:**

1.  Section 2.1 does not appear to provide useful material; it lacks clear connections to the subsequent development, even though useful notation is defined here. For example, what is the purpose of introducing the labeling function $f_\theta$ (or $f_\theta^*)$ and its decomposition $f_\theta = h_w \circ \phi_v$? What do the associated quantities mean, and how do they enter the later development?


  2. In Section 2.2.2, basic symbols such as $b_j$, $w_j$, and $m$ are not defined, which makes the subsequent exposition unclear. Moreover, restricting $\nu$ to a discrete measure with finite support is very limiting and excludes important discrete distributions (e.g., Poisson), which have countably many values.

3. The theorems are not clearly presented. For instance, in Theorem 4,
   are the probability measures
   $\mu_i$ and $\nu_i$
   mean to represent
   $\mu_1,\nu_1,\mu_2,\nu_2$ by letting $i=1,2$?
   How are these measures related to the source and target domains? The optimal transport $T_\nu^\mu$ stated in the conclusion is neither defined nor constructed.
Furthermore, given that Section 2.2.2  defines $\nu$ as a discrete measure with finite support, the necessary conditions for classifying target samples in Theorem 4 are unsurprising. It would be more informative to establish ``necessary and sufficient" conditions for this special finite-support case, or to  consider a more general measure $\nu$.


4. Section 4 aims to extend OT scores to multiclass settings, where the authors state "we model the source distribution in the feature space as a discrete measure and treat the target data as samples drawn from a continuous measure." This does not align with Section 3,
      where the authors declare "For clarity and tractability, we
focus on binary classification tasks.
       Why is such a substantially different scenario introduced? How difficult is it to adapt the Section 3 development, and what would intrinsic challenges arise? These issues are not addressed.


5. The cost $c(x,y)=\|x-y\|^2$ is used interchangeably with the Euclidean distance, which could lead to errors in derived inequalities.

6. The derivation mixes primal and dual OT formulations and fails to specify whether the score is computed using potential gaps or cost differences.


7. In Algorithm 1, the procedure for selecting the learning rate $\gamma$ and the entropic regularization parameter $\epsilon$ is not described; the stopping criterion is also unspecified. It is unclear how different choices of the iteration number max\_iter affect the final OT scores. In addition, the impact of small perturbations in pseudo-labels on OT potentials and scores is not discussed.

8.  Some tables omit standard deviations, making comparisons with benchmark methods incomplete and potentially unfair. A method with higher point accuracy but high variability is not necessarily superior to one with slightly lower accuracy and much smaller variance.

---

### Official Review · Reviewer_UeF4 · 2025-11-10

**Soundness:** 2
**Presentation:** 2
**Contribution:** 2
**Rating:** 4
**Confidence:** 2

**Summary:**

This paper proposes the OT Score, a confidence metric for Source-Free Domain Adaptation (SFDA), built upon Semi-Discrete Optimal Transport (OT) theory. The authors argue that OT Score quantifies the reliability of pseudo-labels by measuring how well a target sample aligns with its assigned source class under the OT-induced geometric structure. They claim that the score is theoretically principled, enables effective pseudo-label reweighting during training, and serves as a label-free proxy for target-domain performance. Experiments on Office-Home, VisDA-17, and ImageCLEF-DA demonstrate improvements in AURC and classification accuracy over several confidence baselines.

**Strengths:**

1. Novel theoretical perspective: This paper provides a fresh theoretical angle on the SFDA problem. The attempt to connect OT geometry with confidence estimation is conceptually interesting and offers a more principled alternative to heuristic uncertainty measures.

2. Practical utility: The proposed OT Score targets two key challenges in SFDA, pseudo-label reliability and model selection, without requiring target labels, which makes it practically useful.

**Weaknesses:**

1. The paper is somewhat difficult to follow and shows a few inconsistencies. In particular, the algorithm relies on source-domain statistics (class-wise mean features; Algorithm 1, Line 1), which partially departs from a strict SFDA setting where only a frozen source model is available and no source data or statistics can be accessed.

2. The experimental section could be further enriched. While the AURC comparisons across multiple datasets are supportive, the improvements in target-domain accuracy and the overall utility of OT Score could be analyzed in more depth (e.g., ablation on reweighting, sensitivity to pseudo-label quality).

**Questions:**

1. In a strict SFDA scenario where source data is inaccessible, how are the source class-wise mean features obtained? Does this require storing additional statistics beyond the source model checkpoint, and if so, how does this align with the source-free claim?

2. Theorem 6 is developed under the assumption of having the correct target partition, whereas in practice only pseudo-labels are available. How robust is OT Score to early-stage pseudo-label noise, which is common in SFDA?

3. DomainNet is a widely adopted benchmark for large-scale SFDA. What is the reason for omitting it from the experiments, and would the method scale to that setting?

---

### Note · Authors · 2025-11-24

**Comment:**

We thank the reviewers, AC, and program committee for their time and constructive feedback. After careful consideration, we decided to withdraw our paper.

**Withdrawal Confirmation:**

I have read and agree with the venue's withdrawal policy on behalf of myself and my co-authors.